# Effect of Binding and Dispersion Behavior of High-Entropy Alloy (HEA) Powders on the Microstructure and Mechanical Properties in a Novel HEA/Diamond Composite

**DOI:** 10.3390/e20120924

**Published:** 2018-12-04

**Authors:** Mingyang Zhang, Wei Zhang, Fangzhou Liu, Yingbo Peng, Songhao Hu, Yong Liu

**Affiliations:** 1Powder Metallurgy Research Institute, Central South University, Changsha 410083, China; 2College of Engineering, Nanjing Agricultural University, Nanjing 210031, China; 3Henan Huanghe Whirlwind Co., Ltd., Xuchang 461500, China

**Keywords:** high-entropy alloy, diamond, composite, powder metallurgy, spark plasma sintering

## Abstract

This study reports the results of the addition of diamonds in the sintering process of a FCC-structured CoCrFeNiMo high-entropy alloy. The effect of raw powder states such as elemental mixed (EM) powder, gas atomization (GA) powder and mechanical alloying (MA) powder on the uniformity of constituent phase was also investigated. Examination of microstructure and evaluation of mechanical properties of the composites depending on the mixing processes were performed. As a result, GA+MA powder composite showed the highest mechanical properties. The experimental results indicated that the powder manufacturing method was an essential parameter to determine the quality of HEA/diamond composites such as the uniformity of phase and binding behavior.

## 1. Introduction

Nowadays, with the rapid development of advanced manufacturing techniques, especially in aerospace, photovoltaics, electronics and communications, there is growing demand for diamond tools [1,2]. Traditionally, diamond tool materials were prepared either by cold-pressing or pressing-and-sintering of a combination of single metal and pre-alloyed powders together with a matrix material, considering the poor thermal stability of diamond and limitations of molding processes, and it is difficult to guarantee the stability and consistency of performance of the matrix materials [3,4]. A new class of materials known as high-entropy alloys (HEAs) have attracted considerable interest by virtue of their outstanding mechanical properties such as high strength, wear and corrosion resistance [5,6,7,8,9]. Unlike a traditional multi-component matrix material, a HEA has a simple crystal structure without any complex phases. HEAs are expected to be adopted increasingly in diamond tools, owing to their impressive features with regard to stability of composition, microstructure, and mechanical properties. FeCoCrNi is considered as one of the most stable HEA alloys [10]. It can be a good matrix material to facilitate the study of the microstructure interface between the HEA matrix and diamond and the mechanical properties of the resulting HEA/diamond composites [11].

At present, the methods of preparing high-entropy alloy powders are mainly gas atomization (GA) and mechanical alloying (MA), and there are also some elemental mixed (EM) methods [12,13,14]. Different preparation methods have different effects on the shape, particle size and composition uniformity of the HEA powders. The purpose of this study is to investigate the effect of powder state (GA, GA + MA and EM + MA) on the microstructure and mechanical properties of HEA/diamond composites, so as to improve the strength, hardness and wear resistance of the composites and enhance the service life of diamond tool materials.

## 2. Materials and Methods

The investigated HEA matrix of FeCoCrNiMo with a nominal composition (Fe_24.1_Co_24.1_Cr_24.1_Ni_24.1_Mo_3.6_ in wt.% was prepared using powder metallurgy. FeCoCrNiMo alloy powders were prepared by gas atomization, and were labeled as GA powders. The GA powders were mechanically milled using conventional planetary ball-milling equipment (the ratio of ball/powders was 10:1, 300 r/min, 20 h), the powders obtained by gas atomization and ball milling were labeled as GA + MA powders. Ball milling powders with five elemental powder mixtures were prepared under the same ball milling conditions, which were labeled as EM+MA powders, as shown in Figure 1. Then these were mixed with 4% (wt.%) synthetic diamond particles (140 mesh) by using a cylinder mixer for 6–8 h. The mixed HEA/diamond powders were then placed in a graphite die of 40-mm diameter and consolidated by using an HPD 25/3 SPS equipment under vacuum (10^−3^ Pa). The SPS process lasted for 20 min, of which the heating time was 10 min, the holding time was 8 min, and the cooling time was 2 min. The sintering temperatures was 950 °C, and the specific pressure was 35 MPa which was selected as SPS parameters for FeCoCrNiMo HEA/diamond composite sintering temperature. Under these conditions, the density, microstructure and properties of the HEA/diamond composite achieved the optimal state (unpublished results). A scanning electron microscope (SEM, FEI, Quanta 250 FEG, Vlastimila Pecha, Czech Republic) equipped with an energy dispersive X-ray (EDX) analyzer was used to investigate the microstructure and chemical compositions of the sintered samples. The phase constitution of the samples was characterized using X-ray diffractometer (XRD, PIGAKV, Rigaku D/MAX-2550 VB+18 Kw, Tokyo, Japan) utilizing Cu Kα radiation. Differential scanning calorimetry (DSC, Netzsh 449C, Selb, Germany) measurements were performed on sintered samples placed in an Al_2_O_3_ crucible and heated. The bending strength of the samples (size: 12×2×30 mm^3^) was determined by Instron 3369 mechanical testing facility (Instron, Norwood, MA, USA) using the three-point method. The hardness of the alloy was determined using a Buehler 5104 hardness tester (Buehler, Lake Bluff, USA) under a 200-g load for 15 s and was averaged from three measurements. The wearing behavior was measured by HRS-2M high-speed reciprocating line friction test equipment (Lanzhou Zhongke Kaihua Technology Development Co., Ltd., Lanzhou, China). The test parameters were a test time of 15 min, 50N loading, 15 Hz frequency (900 times/min), and 5 mm stroke.

## 3. Results and Discussion

### 3.1. Characterization of the Powders

The three different powders were characterized by SEM and XRD. It can be seen in Figure 1a, the GA powders were nearly spherical and had a large particle size distribution. The average particle diameter is about 45–50 μm. After 20 h of ball milling, the particle size of GA + MA powders and EA + MA powders were smaller than that before ball milling as shown in Figure 1b,c. Both of them became flake or strip shaped after ball milling, the particle size distribution of the powders decreased, and the previously larger spherical particles of GA powders were crushed to form fine particles. The powders with different particle size distribution will form different pore size after sintering [15], thus affecting its sintering densification. The particle size of powders has a great influence on the microstructure and mechanical properties. Ball milling is often used to reduce the particle size in order to speed up the diffusion reaction rate and obtain uniform and fine grains. 

From the XRD results of Figure 2a, it can be seen that there was no alloying of GA + MA powders in the milling process and a face-centered-cubic structure (FCC) was maintained. However, the diffraction peaks of the powders were widened and the intensity was obviously reduced after ball milling, indicating that the grain size of the powders was reduced compared with GA powders. Because EM + MA powders were made of element powder by ball milling, mechanical alloying of different metal elements will occur under the action of higher energy in ball milling, resulting in the Cr-rich and Mo-rich area segregation. Moreover, as can be seen from Table 1, EA + MA powders exhibited a great deal of compositional inhomogeneity, especially of the element Cr, which would give rise to large amount of impurity phase after sintering. As shown in Figure 2b, for GA powders, the phase composition after SPS with and without ball milling was basically the same, mainly composed of FCC phase and diamond phase, indicating that there is no impurity formation in the ball milling process, corresponding to Figure 2a. However, for EM + MA powders, due to the inhomogeneity of the powder composition, metastable phases appeared after SPS, like a Cr-rich phase and a Mo-rich phase which were hard and brittle [16]. 

### 3.2. Microstructure of HEA/Diamond Composites

Figure 3(a1–c1) show the microstructure of HEA matrix SPSed by GA, GA + MA, and EM + MA powders, respectively. It can be seen that the GA powder SPS sample exhibited a large number of holes left after sintering, and the densification was poor. The density of GA and EM powders increased significantly after ball milling.

The average particle size of the powders after ball milling (GA + MA and EM + MA) was smaller than that of the GA powders, the voids between the particles are smaller, and the voids are easy to close in the densification stage during sintering. The larger the specific surface area of the particles are, the greater the sintering driving force is, so the GA + MA and EM + MA powders are easier to achieve sintering densification than the GA powders. 

On the other hand, compactness is lower when the powder has a spherical shape than the irregular one after milling. The densification degree also accounts for the green density, prior to hot consolidation stage. The densification parameter is defined as the equation below [17]:Densification parameter = (Green density − Apparent density)/(Theoretical density − Apparent density)

The theoretic density is 8.9 g/cm^3^ for the HEA/diamond composite. The Green density is 7.32, 7.48 and 7.54 g/cm^3^ for GA, GA + MA and EM + MA samples respectively. The apparent density is 4.12, 1.04 and 1.42 g/cm^3^ for GA, GA + MA and EM + MA samples respectively. By calculation, the densification parameters are 0.67, 0.81 and 0.82 for the GA, GA + MA and EM + MA samples respectively. It shows that the densification degree of the samples with ball milling is greater than that without ball milling.

The EBSD images in Figure 3(a2–c2) show the element distribution of HEA matrix SPSed by the three kinds of powder. After GA powders and GA + MA powders were sintered, the matrix composition of the HEA/diamond composites was more evenly distributed. However, for EM + MA powders, Cr-rich and Mo-rich phases were bound to occur in the SPS process, according to the XRD pattern in Figure 2. Because elemental segregations of Cr and Mo were generated in the milling process, the components of Mo and Cr elements fluctuated greatly as seen in Figure 3(c2).The ball milling process fines the powder and introduces lattice distortion into the powder particles. The particle size decreases and the surface energy increases, and the repeated deformation of the powders during ball milling leads to a large number of lattice distortions in the particles after ball milling. On one hand, the decrease of particle size increases the surface energy, on the other hand, it shortens the diffusion path, which leads to the rapid completion of the reaction. Moreover, lattice distortion promotes the diffusion process [18], so that the sintering reaction can be completed at a lower temperature. 

Figure 4a–c show the microstructures of HEA/diamond composite SPSed by GA, GA + MA and EM + MA powders. After the three kind of powders were sintered, the distribution of diamond particles in the composites was relatively uniform, but in the single diamond particle enlargement images shown in Figure 4d, the diamond grains maintain a well-defined crystalline shape and a complete crystal plane. The HEA matrix SPSed by GA powders were found to be closely packed with diamond. For the GA + MA sample, shown in Figure 4e, the diamond particles became partly spherical as the regular diamond particles were ablated into smaller spheres by the molten HEA matrix. For EM + MA sample, obvious spheroidization occurred in the diamond grains, that was attributed to the graphitization transformation of diamond particles. The sintering pressure, temperature and time of the three kind of powders are the same, but the diamond particles in the composites are graphitized. This is because, by means of mechanical ball milling, the distortion energy is introduced while the powder particles are refined, the reaction temperature is reduced and the reaction time is shortened.

### 3.3. Mechanical Properties of HEA/Diamond Composites

#### 3.3.1. Hardness and Bending Strength

Generally, the higher the density of the composites is, the higher the hardness is. The voids between particles of the MA powders are smaller, the voids are easy to be eliminated in the densification stage, and the sintering densification is easier to be realized. As shown in Figure 5, due to the smallest particle size after ball milling, the density of the composite prepared by EM + MA powders is the highest, and the sample prepared by GA powders is the lowest. However, the hardness of EM + MA sample decreases because of the inhomogeneous composition distribution. Considering hardness and density, the GA + MA sample is the best. 

According to the bending stress-strain curves in Figure 6, the bending strength of GA + MA sample is 988 MPa, which is much higher than that of GA and EM + MA samples. The bending strength of GA sample is low is due to the microvoids. The bonding between diamond and matrix is poor, and there is no good metallurgical bonding. There is only a metallurgical bonding and mechanical bonding interface. The bending strength of EM + MA sample is very poor, because of diamond as a hard phase had undergone graphitization and the generation of the segregations like Cr-rich phase and Mo-rich phase. In addition, the elastic slope is different in the curves of the SPSed composites. This phenomenon results from the different density and segregation of samples SPS by different powders. For GA powder, the density is relatively low, microporosity and in HEA matrix results in lower elastic modulus. For EA + MA powders, despite the highest density, a large number of segregation occurred during SPS results in a decline of elastic modulus. GA + MA specimen has relatively higher density and no obvious segregation, so the elastic modulus is high and shows the optimal mechanical properties.

#### 3.3.2. Wear Properties

The basic type of wear is the combination of abrasive wear and adhesive wear [19]. Generally speaking, the friction failure of diamond tools is diamond shedding or matrix wear. According to the wear mechanism, it can be inferred from Figure 7a,c that the abrasive wear was mainly attributed to abrasive wear and was accompanied by a part of adhesive wear because of the grooves and a small amount of abrasive debris on the surface. However, for GA samples, as shown in Figure 7a, there was an obvious interface between diamond particles and HEA matrix, and signs of breaking off were exhibited. This is because of the metallurgical bonding between diamond and HEA matrix is insufficient, and the loose structure and insufficient bonding will cause the diamond particles to break off from the HEA matrix and fail if friction continues. For the EM + MA sample, as shown in Figure 7c, the surfaces of diamond particles were markedly scratched and deformed after friction, which indicates that the diamond particles are graphitized and softened during sintering, thus losing their wear resistance as abrasive particles. For EM + MA sample, as shown in Figure 7b, although the diamond particles were worn, it had no sign of breaking off and softening. This illustrates that the HEA matrix has enhanced the grinding force to diamond particles, and improved the bonding strength of the interface, thus increasing the service life of the HEA/diamond composite. It is also observed from the wear scratch contour map shown in Figure 8 that the composite sample SPSed by GA + MA powders exhibited the minimum depth of wear scratches.

Table 2 shows the wear rate of the HEA/diamond composites SPSed by different powders. The holding force of HEA matrix on diamond particles directly determines the performance of the composite. The interfacial bonding strength between diamond and HEA matrix is the key factor affecting the holding force.

Ball milling can reduce the sintering reaction temperature, improve the alloying degree and densification of the composites, and thus greatly improve the matrix holding force of diamond particles. It can be seen from Table 2 that the wear rate of GA sample is 0.27 mg/min, almost seven times higher than that of GA + MA sample which exhibits minimum worn quality loss. The wear rate of EM + MA sample is higher than GA + MA sample, because graphitization reduces the properties of diamond particles to a certain extent.

As for the friction coefficient, as illustrated in Figure 9, GA + MA and EM + MA samples have similar frictional behavior characteristics: in the first four seconds, the friction coefficient shows a decline (from about 0.4 to about 0.12), and maintains a stable state (between 0.11 and 0.12) afterward. The friction behavior of GA sample is completely different from that of the other two samples. It enters a stable state at the beginning and the friction coefficient is about 0.05. This is mainly due to the integrity of diamond particles in GA samples, and the friction force mainly comes from the diamond particles, so the friction coefficient is always stable. GA + MA and EM + MA samples were ball milled to reduce the reaction temperature and time, which resulted in an interfacial reaction between diamond and HEA matrix. The HEA matrix is involved in friction. Therefore, the samples of GA + MA and EM + MA undergo a relatively long unstable period in the initial stage of friction. At the same time, the holding force of diamond matrix is improved because of the interfacial reaction, and the friction coefficient increases. 

## 4. Conclusions

The phases of GA powders before and after ball milling are basically the same, but the grain size of GA + MA powders decreases significantly. Due to the inhomogeneous distribution of elementary powders, impurity phases occur in EM + MA powders after ball milling. Through the ball milling process, the powder particles are refined and the distortion energy is introduced. The voids between the particles are smaller and easy to close in the densification stage during sintering. The reaction temperature is reduced and the sample is easily densified. The HEA matrix SPSed by GA and GA + MA powders were found to be closely packed with diamond. For the EM + MA samples, graphitization transformation of diamond particles occurred due to the reduction of the reaction temperature. The HEA/diamond composite SPS by GA + MA powders has the optimal hardness, bending strength and wear resistance. Although the EM + MA sample is more dense, the mechanical properties decrease due to the generation of segregations like Cr-rich phase and Mo-rich phase.

## Figures and Tables

**Figure 1 entropy-20-00924-f001:**
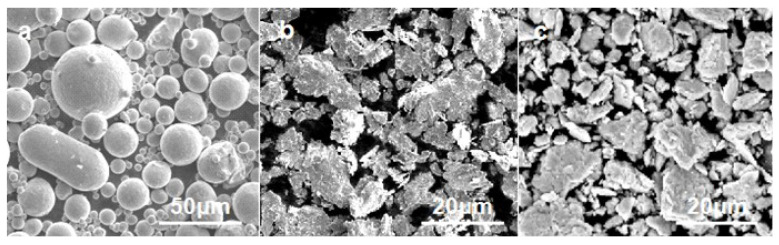
FeCoCrNiMo HEA powders (**a**) GA; (**b**) GA + MA; (**c**) EM + MA.

**Figure 2 entropy-20-00924-f002:**
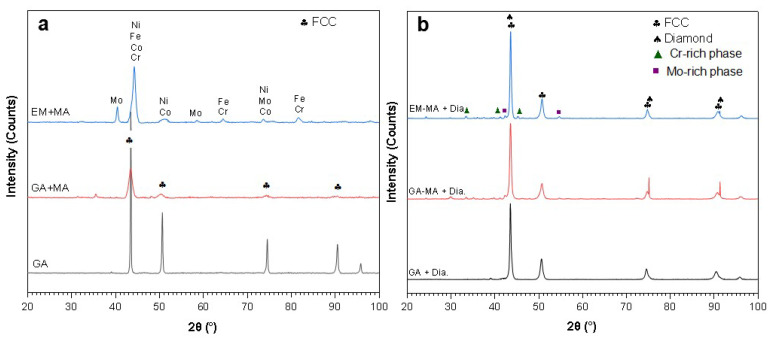
XRD pattern of (**a**) Three kind of FeCoCrNiMo HEA powders; (**b**) HEA/diamond composites after SPS by different powders.

**Figure 3 entropy-20-00924-f003:**
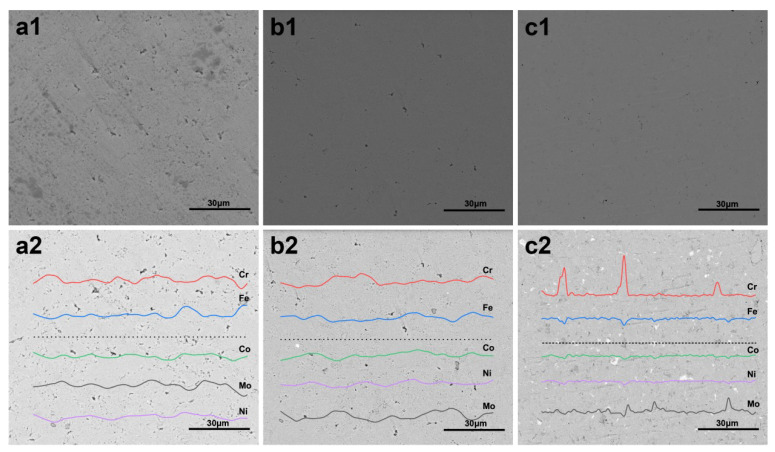
Microstructure (**a1**–**c1**) and element distribution (**a2**–**c2**) of HEA matrix SPSed by (**a**) GA powders, (**b**) GA + MA powders, and (**c**) EM + MA powders.

**Figure 4 entropy-20-00924-f004:**
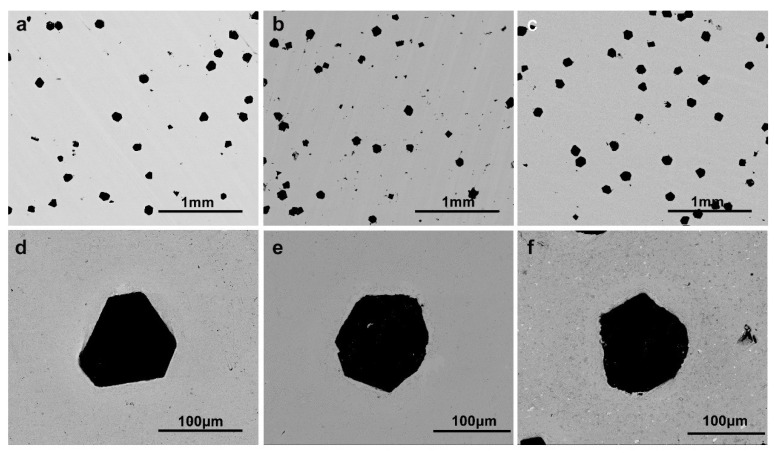
Microstructure of HEA/diamond composite SPSed by (**a**) and (**d**) GA powders; (**b**) and (**e**) GA + MA powders; and (**c**) and (**f**) EM + MA powders.

**Figure 5 entropy-20-00924-f005:**
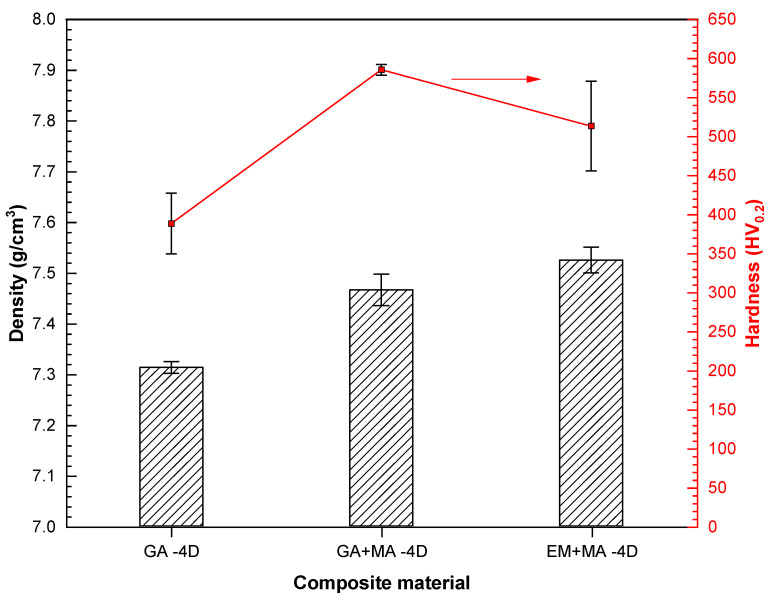
Hardness and density of HEA/diamond composites SPSed by three kinds of powders.

**Figure 6 entropy-20-00924-f006:**
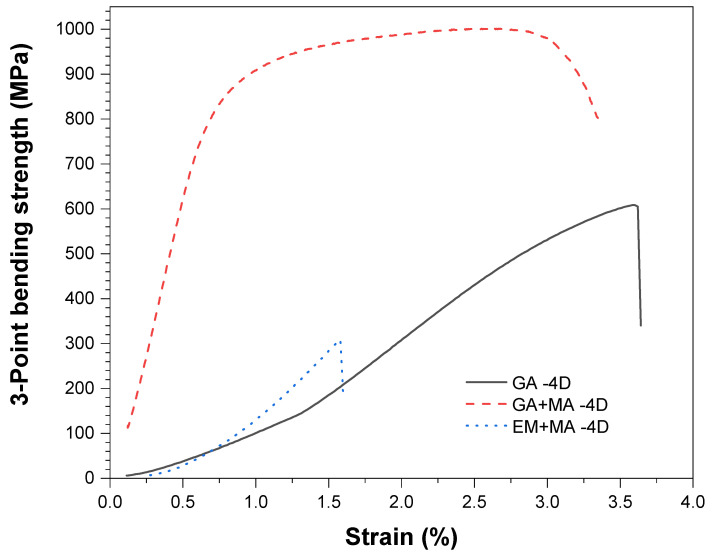
Bending stress-strain curves of HEA/diamond composites SPSed by three kinds of powders.

**Figure 7 entropy-20-00924-f007:**
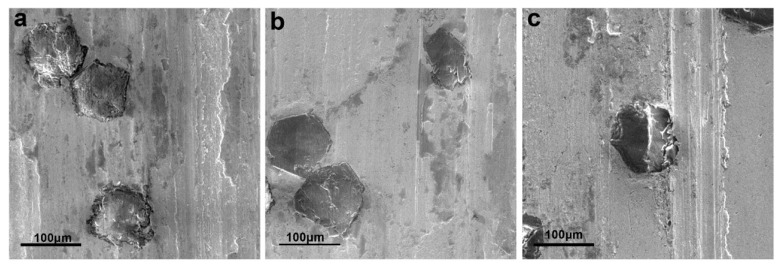
The wearing surface of HEA/diamond composite SPSed by (**a**) GA powders; (**b**) GA + MA powders; and (**c**) EM + MA powders.

**Figure 8 entropy-20-00924-f008:**
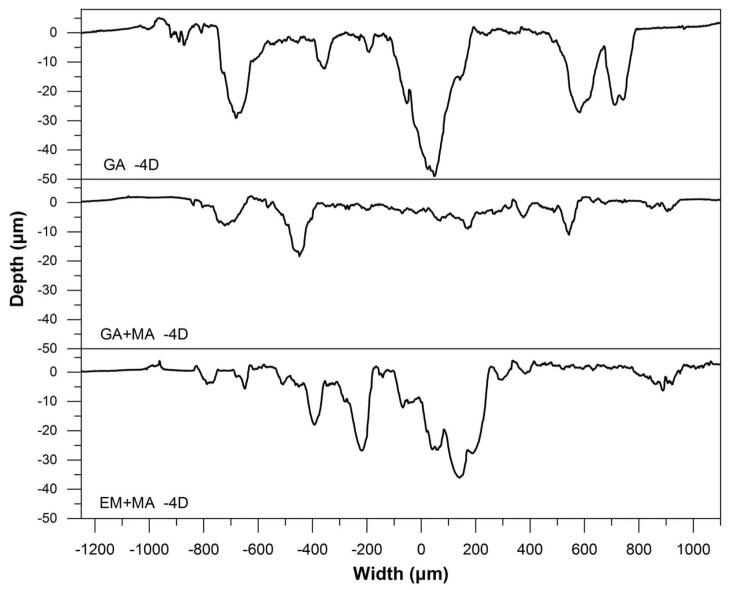
Wear scratch contour map of HEA/diamond composites SPSed by three kinds of powders.

**Figure 9 entropy-20-00924-f009:**
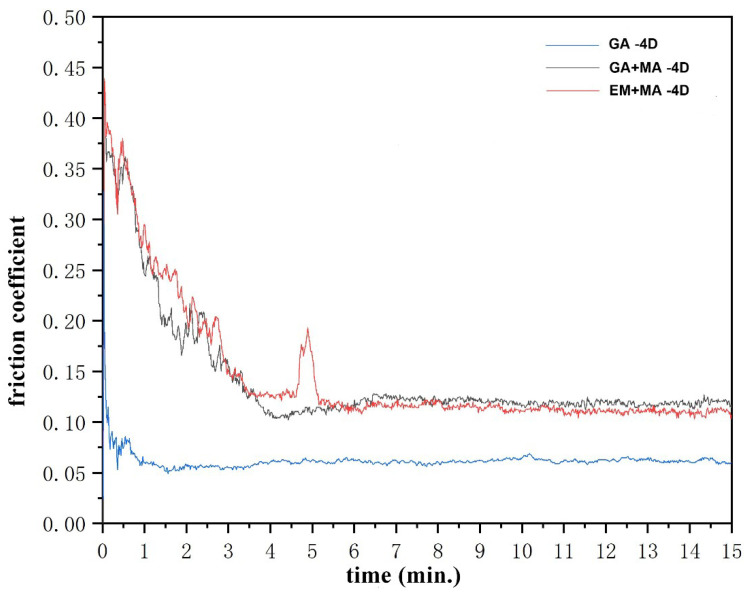
Friction coefficient-time curves of HEA/diamond composites SPSed by three kinds of powders.

**Table 1 entropy-20-00924-t001:** Composition of GA, GA + MA and EM + MA powders.

Elements	Fe	Co	Cr	Ni	Mo
Nominal composition (at. %)	24.10	24.10	24.10	24.10	3.6
GA powders	23.83 ± 0.15	22.89 ± 0.23	30.31 ± 0.34	18.53 ± 0.45	4.44 ± 0.23
GA-MA powders	26.70 ± 0.75	22.41 ± 0.50	29.07 ± 0.32	18.13 ± 0.34	3.68 ± 0.67
EM-MA powders	27.17 ± 14.13	15.41 ± 11.10	40.33 ± 29.17	14.07 ± 10.42	3.03 ± 2.00

**Table 2 entropy-20-00924-t002:** Wear rate of the HEA/diamond composites SPSed by different powders.

Composites	Wear Rate (mg/min)
GA-dia	0.27
(GA + MA)-dia	0.04
(EM + MA)-dia	0.06

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
