# Peer review of "Effect of Binding and Dispersion Behavior of High-Entropy Alloy (HEA) Powders on the Microstructure and Mechanical Properties in a Novel HEA/Diamond Composite"

_entropy, 2018, doi:10.3390/e20120924_

Round 1
Reviewer 1 Report
The manuscript shows quite comprechensive study of HEA-diamond composites prepared by SPS. Mechanical properties of composites show quite significant improvement in a comparison with peren. Potentially, results has importnace for the community.
line 58. choose of optimal conditions is not addressed. I am not sure that it is enought to refere to unpublished results. Probably short section should be added to give an idea about parameters varied.
figure 2-b-b. the origine of high 2theta sharp and narrow difraction lines is not clear as well as low 2theta lines.
I suggest to give more references such as Riva et al. (2018) J. Alloys Comp. 730 p544-551
Author Response
Dear Reviewer 1,
Thank you very much for your valuable and constructive suggestions and comments concerning our manuscript entitled “Effect of binding and dispersion behavior of high-entropy alloy (HEA) powders on microstructure and mechanical properties in a novel HEA/diamond composite”. Those comments are truly valuable and helpful for enhancing the quality of our paper, as well as the important guiding significance to our research. We have studied all comments carefully and have made substantially corrections of our manuscript. With these revisions, we believe that our revised paper is satisfactory to be accepted for publication in Entropy. The point-by-point responses to the comments and suggestions are given in details below.
Comment1: Line 58. choose of optimal conditions is not addressed. I am not sure that it is enough to refer to unpublished results. Probably short section should be added to give an idea about parameters varied.
Response1:We would like to thank Reviewer #1 for your valuable suggestion and have already added some brief explanation about parameters about SPS. In fact, we have done systematic experiments, and obtained the parameter of pressure and temperature in SPS, with the optimal microstructures and properties of the specimens.
The change was made in P. 2:
The sintering temperatures was 950 °C, and the specific pressure was 35 MPa which was selected as SPS parameters for FeCoCrNiMo HEA/diamond composite sintering. Under this condition, the density, microstructure and properties of the HEA/diamond composite achieve the optimal state (unpublished results).
Comment2: Figure 2-b-b. the origine of high 2theta sharp and narrow difraction lines is not clear as well as low 2theta lines.
Response2:We are sorry that Fig. 2 about the XRD of HEA/diamond composite in the manuscript not clear. We have already replaced Fig. 2 with clear pictures.
The change was made in P. 3:
Fig. 2 XRD pattern of (a) Three kind of FeCoCrNiMo HEA powders; (b) HEA/diamond composites after SPS by different powders.
Comment3: I suggest to give more references such as Riva et al. (2018) J. Alloys Comp. 730 p544-551.
Response3:Thank the reviewer #1’s kindly remind. We have carefully read and quoted the references as mentioned.
Please let us know if any additional information is needed. We are looking forward to receiving a positive decision on our revised manuscript for publication in Entropy.
Sincerely yours,
Reviewer 2 Report
Reviewer’s Comments on manuscript Entropy-388182 titled “Effect of binding and dispersion behavior of high-entropy alloy (HEA) powders on microstructure and mechanical properties in a novel HEA/diamond composite”.
The authors have used elemental mixing, gas atomization and mechanical alloying techniques to assess the physical properties of a high entropy alloy (CoCrFeNiMo) with the addition of diamond.
1) The manuscript needs major revision in terms of language and sentence formation.
2) Why is that particular composition chosen, any particular advantage, why is Mo added? No references provided on the past work on this composition.
3) Lines 114-116: EM+MA powders, Cr-rich, and Mo-rich phases were bound to occur in the SPS process because impurity phases like μ and σ phases were generated in the milling process, according to the XRD. What do the authors mean by impurity phases? These are not carbides or oxides?
4) Many peaks in XRD data (Figure 2) are not indexed!
5) Higher magnification SEM images of μ and σ phases should be provided along with EDS maps.
6) No proof of graphitization of the diamond particles!
7) No quantitative wear data is provided. The surface images are insufficient to comment on the wear resistance.
The manuscript needs a major revision and more detailed analysis for publication. The description is based on assumptions and incomplete analyses.
Author Response
DearReviewer 2,
Thank you very much for your valuable and constructive suggestions and comments concerning our manuscript entitled “Effect of binding and dispersion behavior of high-entropy alloy (HEA) powders on microstructure and mechanical properties in a novel HEA/diamond composite”. Those comments are truly valuable and helpful for enhancing the quality of our paper, as well as the important guiding significance to our research. We have studied all comments carefully and have made substantially corrections of our manuscript. With these revisions, we believe that our revised paper is satisfactory to be accepted for publication in Entropy. The point-by-point responses to the comments and suggestions are given in details below.
Comment1: Why are data from EM powders not given in the manuscript? There is no evaluation of the influence of milling when the alloys/composites are processed from elemental powders.
Response1:Thanks for Reviewer #2’ comment. In fact, elemental powders (EM) are not usually be used in SPS without ball milling. The elemental powders can be pre-alloyed by ball milling (mechanical alloying) and the composition uniformity of the powder can be improved as far as possible. Mechanical alloying is not simply mixing on a fine scale: true alloying occurs (G. S. Upadhyaya. Powder Metallurgy Technology [C]. Cambridge International Science Publishing. 2012, p36). In our manuscript, GA powder is a powder that has been alloyed. But EM powder still need pre-alloying by ball milling.
Comment2: Lines 73-74. It is written that "particle size of GA + MA powders and EA + MA powders were smaller than that before ball milling". This is not obvious in the images presented. In fact, micrographs evidence a relatively compact material in which the particle size cannot be discriminated, as expected from the repeatedly breaking/welding processes occurring during the mechanical grinding. Only the grain size will be much finer because of the large deformation induced in the particles during the milling stage.
Response2:We would like to thank Reviewer #2 for your valuable suggestion and replaced SEM images of GA+MA powders and EA+MA powders with higher magnification. So, the difference between the particle sizes of the three powders can be clearly observed.
The change was made in P. 3:
Fig. 1 FeCoCrNiMo HEA powders (a) GA; (b) GA+MA; (c) EM+MA
Comment3: Lines 106-110. The authors state that "the larger the specific surface area of the particles are, the greater the sintering driving force is, so the GA+MA and EM+MA powders are easier to achieve sintering densification than the GA powders". Although this statement is true, the morphology of the powder also accounts for the compactness of green compacts, prior to hot consolidation stage. As a rule, compactness is lower when the powder has a spherical shape than the irregular one after milling. Consequently, the higher densification degree achieved in GA+MA and EM+MA cannot be exclusively attributed to the higher specific surface area.
Response3:We are really admire and admit Reviewer #2’s accurate description of factors affecting the densification degree. The compactness of green compacts is indeed a major factor in powder metallurgy. We measured the compactness of green compacts of three kinds of powders and illustrated in the manuscript.
The change was made in P. 4:
On the other hand, compactness is lower when the powder has a spherical shape than the irregular one after milling. The densification degree also accounts for the green density, prior to hot consolidation stage. The densification parameter is defined as the equation below:
Densification parameter=(Green density - Apparent density)/ (Theoretic density - Apparent density)[17]
The theoretic density is 8.9 g/cm3for the HEA/diamond composite. The green density is 7.32, 7.48 and 7.54 g/cm3for GA, GA+MA and EM+MA samples respectively. The apparent density is 4.12, 1.04 and 1.42 g/cm3for GA, GA+MA and EM+MA samples respectively. By calculated, the densification parameter is 0.67, 0.81 and 0.82 for GA, GA+MA and EM+MA samples respectively. It is illustrated that the densification degree of the samples with ball milling is greater than that without ball milling.
Comment4: Lines 130-142. The authors claim changes in the crystallinity of diamond particles depending on the nature of staring powders. However, general view of the composites does not reveal differences between them. Spheroidization of diamond particles in the GA+MA composite is related to ablation produced by the molten HEA matrix. Why does the same process not happen for GA composite? It is not clear how the molten matrix can turn round the hard diamond particles. In the case of EA+MA composites, the rounding shape is explained because of the occurrence of graphitization of diamond particles. Why does not graphitization occur when GA or GA+MA powders are used? Furthermore, has it been checked by calorimetric measurements that reaction temperature is reduced when milled powders are used?
Response4:This is a good question! We accept Reviewer #2’s suggestion and have added the corresponding explanation in the manuscript.
The change was made in P. 5-6:
By molecular dynamics simulation, diamond begins to graphitized at 1200-1700 ℃ in this experiment, much higher than the temperature used in SPS of 950 ℃. According to the DSC examination, it can be seen from the Fig. 5, for GA+MA powders, due to the large number of lattice distortion and surface energy introduced by ball milling, the reaction temperature in SPS is significantly reduced to 727 ℃, no graphitization of diamond particles will happen and high sintering densification is easy to achieve. For GA powders, the reaction temperature was 881 ℃ without ball milling. Although the densification is low, graphitization will not occur.
Fig. 5DSC of HEA/diamond composites SPSed by three kinds of powders
However, for EA+MA powders, although the sintering density is the highest, the reaction temperature reaches 884 ℃. Because of the alloying degree of EM+MA powders are relatively low, there will be two effects on sintering of HEA/diamond composite. Firstly, due to the low alloying degree, fewer phases which have low solid phase transformation point (FCC or BCC phases in HEA) are formed, so the reaction temperature in SPS can not be reduced. Secondly, unalloyed elemental metals (Fe, Co, Cr, Ni) in the HEA matrix will react with C at very low temperatures to form carbides (as known from the phase diagrams of Fe-C and Cr-C, such as 727 ℃ in Fe-C, 627 ℃ in Cr-C), resulting in significant graphitization of diamonds in EM+MA specimen during SPS.
Comment5: Why is the elastic slope in the curves of the composites different? Very small differences could be expected.
Response5:Firstly, we approve your point of view that in the same materials, the elastic slope in the curves should exhibited little differences. However, we have been carefully tested the HEA/diamond samples again, and the same result is obtained. This phenomenon results from the different density and segregation of samples SPS by different powders. For GA powder, the density is relatively low, microporosity and in HEA matrix results in lower elastic modulus. For EA+MA powders, despite the highest density, a large number of segregation occurred during SPS results in a decline of elastic modulus. GA+MA specimen has relatively higher density and no obvious segregation, so the elastic modulus is high and shows the optimal mechanical properties.
Comment6: In general, the description and explanation of features associated with the abrasive process should be improved. Otherwise wear study is very weak and simple. Wear rate curves should be provided and, then, these results correlated with the microstructure of the alloys.
Response6:We would like to thank Reviewer #2’s valuable advice of our work. We totally agree with you that the description and explanation of features associated with the abrasive process should be improved. Friction coefficient curve, wear scratch profile curve and wear coefficient are added in the revised manuscript.
The change was made in P. 8-10:
It is also observed from the wear scratch contour map shown in Fig. 9 that the composite sample SPSed by GA+MA powders exhibited the minimum depth of wear scratches.
Fig. 8The wearing surface of HEA/diamond composite SPSed by (a) GA powders; (b) GA+MA powders; and (c) EM+MA powders.
Fig. 9Wear scratch contour map of HEA/diamond composites SPSed by three kinds of powders
As for the friction coefficient, as illustrated in Fig. 10, GA+MA and EM+MA samples have similar frictional behavior characteristics: In the first four seconds, the friction coefficient show a declination (from about 0.4 to about 0.12), and maintain a stable state (between 0.11 and 0.12) afterward. The friction behavior of GA sample is completely different from that of the other two samples. It enter a stable state at the beginning and the friction coefficient is about 0.05. This is mainly due to the integrity of diamond particles in GA samples, and the friction force mainly comes from the diamond particles, the friction coefficient is always stable. GA+MA and EM+MA samples were ball milled to reduce the reaction temperature and time, which resulted in the interfacial reaction between diamond and HEA matrix. The HEA matrix is involved in friction. Therefore, the samples of GA+MA and EM+MA undergo a relatively long unstable period in the initial stage of friction. At the same time, the holding force of diamond matrix is improved because of the interfacial reaction, and the friction coefficient increases.
Fig. 10Friction coefficient-time curves of HEA/diamond composites SPSed by three kinds of powders
Comment7: Minor points:
1) Line 48. It is written that "Sherical FeCoCrNiMo alloys powders". This reviewer assumes that composition of the powders is FeCoCrNiMo0.15. Correct this in the new version of the manuscript.
2) Line 56. Time for SPS consolidation should be provided.
3) Line 63. Indicate what kind of hardness was measured (Vickers, Brinell,.Rockwell,...)
Response:Thanks of Reviewer #2 for the valuable suggestions.
1) We have corrected this description in the revised manuscript.
2) Time for SPS consolidation has been provided in the section of Materials and Methods.
3) Vickers hardness has been marked in Figure 6 in the revised manuscript.
Please let us know if any additional information is needed. We are looking forward to receiving a positive decision on our revised manuscript for publication in Entropy.
Sincerely yours,
Reviewer 3 Report
The manuscript deals with the influence of mechanical alloying on the microstructure and properties of HEA-based composites reinforced with diamond particles. The study is interesting, especially regarding applications in which high wear resistance is required. Nevertheless, the manuscript should be improved before it can be accepted for publishing. Here is a list of point that should be addressed in a new version of the manuscript.
1) Why are data from EM powders not given in the manuscript? There is no evaluation of the influence of milling when the alloys/composites are processed from elemental powders.
2) Lines 73-74. It is written that "particle size of GA + MA powders and EA + MA powders were smaller than that before ball milling". This is not obvious in the images presented. In fact, micrographs evidence a relatively compact material in which the particle size cannot be discriminated, as expected from the repeatedly breaking/welding processes occurring during the mechanical grinding. Only the grain size will be much finer because of the large deformation induced in the particles during the milling stage.
3) Lines 106-110. The authors state that "the larger the specific surface area of the particles are, the greater the sintering driving force is, so the GA+MA and EM+MA powders are easier to achieve sintering densification than the GA powders". Although this statement is true, the morphology of the powder also accounts for the compactness of green compacts, prior to hot consolidation stage. As a rule, compactness is lower when the powder has a spherical shape than the irregular one after milling. Consequently, the higher densification degree achieved in GA+MA and EM+MA cannot be exclusively attributed to the higher specific surface area.
4) Lines 130-142. The authors claim changes in the crystallinity of diamond particles depending on the nature of staring powders. However, general view of the composites does not reveal differences between them. Spheroidization of diamond particles in the GA+MA composite is related to ablation produced by the molten HEA matrix. Why does the same process not happen for GA composite? It is not clear how the molten matrix can turn round the hard diamond particles. In the case of EA+MA composites, the rounding shape is explained because of the occurrence of graphitization of diamond particles. Why does not graphitization occur when GA or GA+MA powders are used? Furthermore, has it been checked by calorimetric measurements that reaction temperature is reduced when milled powders are used?
5) Why is the elastic slope in the curves of the composites different? Very small differences could be expected.
6) In general, the description and explanation of features associated with the abrasive process should be improved. Otherwise wear study is very weak and simple. Wear rate curves should be provided and, then, these results correlated with the microstructure of the alloys.
Minor points:
1) Line 48. It is written that "Sherical FeCoCrNiMo alloys powders". This reviewer assumes that composition of the powders is FeCoCrNiMo0.15. Correct this in the new version of the manuscript.
2) Line 56. Time for SPS consolidation should be provided.
3) Line 63. Indicate what kind of hardness was measured (Vickers, Brinell,.Rockwell,...)
Author Response
DearReviewer 3,
Thank you very much for your valuable and constructive suggestions and comments concerning our manuscript entitled “Effect of binding and dispersion behavior of high-entropy alloy (HEA) powders on microstructure and mechanical properties in a novel HEA/diamond composite”. Those comments are truly valuable and helpful for enhancing the quality of our paper, as well as the important guiding significance to our research. We have studied all comments carefully and have made substantially corrections of our manuscript. With these revisions, we believe that our revised paper is satisfactory to be accepted for publication in Entropy. The point-by-point responses to the comments and suggestions are given in details below.
Comment1: The manuscript needs major revision in terms of language and sentence formation.
Response1:We appreciate the Reviewer #3’s advice and have asked an expert in English to go over it and made some improvements in the revised manuscript.
Comment2: Why is that particular composition chosen, any particular advantage, why is Mo added? No references provided on the past work on this composition.
Response2:We would like to thank Reviewer #3 for your valuable suggestion and we provide some evidence of the past work on this composition in the revised manuscript.
FeCoCrNi is considered as one of most stable alloys of HEA. (B. Gludovatz, A. Hohenwarter, D. Catoor, E. H. Chang, E. P. George, R. O. Ritchie. A fracture-resistant high-entropy alloy for cryogenic applications [J]. Science, 2014, 345 (6201): 1153-1158.)
Mo has a moderately large atomic size for both solid solution and precipitation hardening. Mo is seleted as a promising element added to the CoCrFeNi HEA. (W.H. Liu, Z. P. Lu, J. Y. He, J. H. Luan, Z. J. Wang, et al. Ductile CoCrFeNiMo x high entropy alloys strengthened by hard intermetallic phases [J]. Acta Materialia, 2016, 116: 332-342.)
Comment3: Lines 114-116: EM+MA powders, Cr-rich, and Mo-rich phases were bound to occur in the SPS process because impurity phases like μand σphases were generated in the milling process, according to the XRD. What do the authors mean by impurity phases? These are not carbides or oxides?
Response3:Thanks for the Reviewer #3’s comment and we are sorry that our descriptions about XRD in the manuscript is not clear. We have been carefully examined the XRD pattern and re-indexed the phases. It is inappropriate to call “impurity phase”, and μ and σ phases can not be determined only by XRD pattern. The re-indexed XRD patterns are shown as below (The change was made in P. 3):
Fig. 2 XRD pattern of (a) Three kind of FeCoCrNiMo HEA powders; (b) HEA/diamond composites after SPS by different powders.
Comment4: Many peaks in XRD data (Figure 2) are not indexed.
Response4:We are sorry that our descriptions about XRD in the manuscript is not clear. We have been carefully examined the XRD pattern and re-indexed the phases.
Comment5: Higher magnification SEM images of μ and σ phases should be provided along with EDS maps.
Response5:Thank the reviewer #3’s suggestion. Firstly, we had re-indexed the XRD pattern and we reconsidered that the μ and σ phases can not be determined only by XRD pattern. According to the EBSD image of HEA matrix SPSed by EM+MA powders, as shown in Fig.3(c), the composition of Cr and Mo element fluctuated relatively greater than the other two samples, indicating that Cr-rich and Mo-rich phases were produced, not μ and σ phases. Corresponding modifications have also been made in the revised manuscript (P.3, P. 8 and P .10).
Comment6: No proof of graphitization of the diamond particles!
Response6:Thanks for Reviewer #3’ comment. We have re-indexed the carbide phases in XRD pattern. Moreover, we examined the SEM image and EDX mapping of C and Cr elements at the HEA/diamond interface of composites SPSed by three kind of powders. It can be seen clearly from the figure that, for EM+MA powder sample, the C element diffused to the matrix, and a large number of diffusion reactions occurred. The C element was enriched around the interface and deep into the matrix, and the C element reacted with the Cr element to form chromium carbide as Cr3C2or Cr7C3. That is to say, the graphitization of diamond particles is proved.
SEM image and EDX mapping of C and Cr elements at the HEA/diamond interface SPSed by (a) GA powders, (b) GA+MA powders, (c) EM+MA powders.
Comment7: No quantitative wear data is provided. The surface images are insufficient to comment on the wear resistance.
Response7:We would like to thank Reviewer #3’s valuable advice of our work. We totally agree with you that the description and explanation of features associated with the abrasive process should be improved. We did some supplementary experiments. Friction coefficient curve, wear scratch profile curve and wear coefficient are added in the revised manuscript.
The change was made in P. 8-10:
It is also observed from the wear scratch contour map shown in Fig. 9 that the composite sample SPSed by GA+MA powders exhibited the minimum depth of wear scratches.
Fig. 8The wearing surface of HEA/diamond composite SPSed by (a) GA powders; (b) GA+MA powders; and (c) EM+MA powders.
Fig. 9Wear scratch contour map of HEA/diamond composites SPSed by three kinds of powders
As for the friction coefficient, as illustrated in Fig. 10, GA+MA and EM+MA samples have similar frictional behavior characteristics: In the first four seconds, the friction coefficient show a declination (from about 0.4 to about 0.12), and maintain a stable state (between 0.11 and 0.12) afterward. The friction behavior of GA sample is completely different from that of the other two samples. It enter a stable state at the beginning and the friction coefficient is about 0.05. This is mainly due to the integrity of diamond particles in GA samples, and the friction force mainly comes from the diamond particles, the friction coefficient is always stable. GA+MA and EM+MA samples were ball milled to reduce the reaction temperature and time, which resulted in the interfacial reaction between diamond and HEA matrix. The HEA matrix is involved in friction. Therefore, the samples of GA+MA and EM+MA undergo a relatively long unstable period in the initial stage of friction. At the same time, the holding force of diamond matrix is improved because of the interfacial reaction, and the friction coefficient increases.
Fig. 10Friction coefficient-time curves of HEA/diamond composites SPSed by three kinds of powders
Please let us know if any additional information is needed. We are looking forward to receiving a positive decision on our revised manuscript for publication in Entropy.
Sincerely yours,
Round 2
Reviewer 2 Report
Accept in present form.
Author Response
Thanks for Reviewer #2’s advice and we have described this phenomenon in the revised manuscript.
Reviewer 3 Report
The quality of the manuscript has been considerably improved compared to the original submission. Nevertheless, there are three points that should be considered in a new version of the manuscript. After that the paper will be acceptable for publishing.
1) The authors invoke changes in the crystalline shape of diamond particles. In the low magnification images no such changes are noticed but the authors state such loss of crystallinity based on images from a unique diamond particle. Moreover, there is no correlation between transformations recorded in DSC curves with microstructural changes. What are microstructural changes associated with each peak in the DSC curve? For the GA material not graphitization is occurring while for the EM+MA alloy graphitization is taking place. It has no sense different transformations when one peak appears at 881 °C for the GA material and 884°C for the EM+MA alloy, and this transformation is the same for both materials. This part of the text is confusing and it should be removed from the manuscript.
2) As written in the author's reply document, the different slope of the elastic part of the curves is due to different level of porosity. This should be clearly described in the text and/or the figure caption.
3) A table with the wear rate of the composites should be given. For comparing purposes, it would be also desirable to provide the wear rate of metallic matrices (GA, GA+MA and EM+MA).
Round 3
Reviewer 3 Report
The paper is suitable for publishing after the changes introduced in the text.